# Characterization and Expression Analysis of Mollusk-like Growth Factor: A Secreted Protein Involved in Pacific Abalone Embryonic and Larval Development

**DOI:** 10.3390/biology11101445

**Published:** 2022-10-01

**Authors:** Md Abu Hanif, Shaharior Hossen, Yusin Cho, Zahid Parvez Sukhan, Cheol Young Choi, Kang Hee Kho

**Affiliations:** 1Department of Fisheries Science, Chonnam National University, Yeosu 59626, Korea; 2Division of Marine BioScience, National Korea Maritime and Ocean University, Busan 49112, Korea

**Keywords:** mollusk-like growth factor, Pacific abalone, digestive gland, catalytic activity, larval development

## Abstract

**Simple Summary:**

The Pacific abalone, *Haliotis discus hannai*, is a gastropod mollusk in high demand, which is extensively cultured in many Asian countries. The growth and development of Pacific abalone depend on the activation of growth factors and other growth-regulating proteins. Growth factors are secreted, biologically active molecules that stimulate cell growth through signal transduction pathways. The mollusk-like growth factor (MLGF) is a mollusk specific growth factor in the adenosine deaminase related growth factor subfamily having a conserved adenosine deaminase (ADA) domain. Molecular functions of this growth factor include adenosine deaminase activity, growth factor activity, and zinc binding activity. For this growth factor activity, enzymatic activity (ADA activity) that converts adenosine to inosine to stimulate cell growth is essential. As abalone aquaculture completely depends on hatchery-produced seed, proper embryonic and larval development is essential, and MLGF is one of the main growth factors that can regulate this activity. In Pacific abalone, higher expression of *Hdh-MLGF* mRNA in the embryonic and larval development stages is an indication of higher production of adenosine and increases the growth factor activity that stimulates embryonic and larval cell growth through signal transduction pathways.

**Abstract:**

Growth factors are mostly secreted proteins that play key roles in an organism’s biophysical processes through binding to specific receptors on the cell surface. The mollusk-like growth factor (*MLGF*) is a novel cell signaling protein in the adenosine deaminase-related growth factor (ADGF) subfamily. In this study, the *MLGF* gene was cloned and characterized from the digestive gland tissue of Pacific abalone and designated as *Hdh-MLGF*. The transcribed full-length sequence of *Hdh-MLGF* was 1829 bp long with a 1566 bp open reading frame (ORF) encoding 521 amino acids. The deduced amino acid sequence contained a putative signal peptide and two conserved adenosine deaminase domains responsible for regulating molecular function. Fluorescence in situ hybridization localized *Hdh-MLGF* in the submucosa layer of digestive tubules in the digestive gland. The mRNA expression analysis indicated that *Hdh-MLGF* expression was restricted to the digestive gland in the adult Pacific abalone. However, *Hdh-MLGF* mRNA expressions were observed in all stages of embryonic and larval development, suggesting *Hdh-MLGF* might be involved in the Pacific abalone embryonic and larval development. This is the first study describing *Hdh-MLGF* and its involvement in the Pacific abalone embryonic and larval development.

## 1. Introduction

Growth factors are cell signaling proteins that orchestrate growth and development related complex regulatory network functions [1]. Growth factors regulating cell proliferation and differentiation are extensively distributed in both vertebrate and invertebrate species [2]. In invertebrates, several growth factors with vertebrate homologues have been assigned at the gene level [3], but few vertebrate growth factors homologous to their invertebrate counterparts have been identified. Growth factors can bind specific cell surface receptors, causing conformational changes of the receptors that enable them to transmit specific signals for cell growth [4] by activating signal transduction pathways [5].

The adenosine deaminase-related growth factor (ADGF) family is a group of growth factors with carboxyl-terminal sequence similar to classical adenosine deaminase (ADA) [6]. The members of this ADGF family are secreted proteins having growth factor activity and regions of ADA homology in insects, mollusks, and vertebrates [7]. The ADGF subfamily includes adenosine deaminase (ADA), mollusk-like growth factor (MLGF), mollusk derived growth factor (MDGF), tsetse salivary growth factor (TSGF), insect-derived growth factor (IDGF), and cat eye syndrome critical region (CECR) gene. Some members have secretory or transmembrane-type ADGFs with common ADA activity, whereas others have growth factor-like activities [8]. This ADA activity is indispensable for growth factor activity [2]. Most genes in this growth factor family have mitogenic activity and are thus involved in embryonic and larval development [8,9,10,11,12]. 

Mollusk-like growth factor (MLGF) is a novel gene in the ADGF subfamily having ADA activity and an additional ADA in the N-terminal region. MLGF was first purified from a normalized cDNA library of disc abalone, *Haliotis discus discus*. MLGF is structurally similar to MDGF, a member of the ADGF subfamily specific to molluscan invertebrate that requires enzymatic activity for its growth factor activity, a unique property of known growth factors in the same subfamily [13]. 

Abalones are a group of reef-dwelling marine gastropod mollusks belonging to the snail family, Haliotidae. Approximately 56 species of abalone are found in tropical and temperate coastal reefs and rocky habitats of Australia, the United States and East Asia [14], though most of them inhabit temperate waters. In recent decades, the natural abalone has experienced intensive fishing throughout the world, due to a high market demand, which resulted in natural stock depletion in most countries [15]. As a result, abalone aquaculture was established in South Korea, which has largely expanded to reduce the fishing pressure of natural production. Among the available abalone species (*H. medaka*, *H. discus hannai*, *H. diversicolor*, *H. discus discus*, *H. gigantea*, *H. diversicolor supertexta*), the Pacific abalone, *H. discus hannai*, is a highly valued, commercially important species and is abundantly cultured not only in South Korea [16,17], but also in many Asian countries [18,19]. As Pacific abalone aquaculture completely depends on hatchery-produced seeds [20], the quality of the seed is a vital issue for Pacific abalone aquaculture production from. Embryonic and larval development are essential processes that can affect the quality of seed and growth status as well as determine aquaculture economic performance [21]. The molecular mechanism of an organism’s development is critical and is directly or indirectly regulated by different genes, including different growth factors. In this study, mollusk-like growth factor (MLGF) was cloned and characterized from the digestive gland tissue of pacific abalone, *H. discus hannai*. Henceforth, localization of MLGF was performed using fluorescence in situ hybridization in adult Pacific abalone. Furthermore, qRT-PCR was executed to observed tissue distribution and temporal expression in different organs, and to predict its potential role in embryonic and larval development of Pacific abalone, *H. discus hannai*.

## 2. Materials and Methods

### 2.1. Experimental Animal and Sample Collection

Mature Pacific abalone of both sexes were collected from the abalones sea cage culture area of having a mean body weight of 120.4 ± 0.61 g and mean shell length of 84.06 ± 0.32 mm were collected from sea cages (Wando-gun, Korea). Collected abalone were transported to the molecular physiology laboratory in the Department of Fisheries Science, Chonnam National University, Yeosu, Korea.

### 2.2. Collection of Organs for Gene Cloning and In Situ Hybridization

We randomly selected 15 abalones of both sexes that were anesthetized with 5% MgCl2 and then sacrificed for different tissue collection to perform cloning and isolation of MLGF. Collected digestive gland tissues were then carefully washed with phosphate-buffered saline (PBS, 0.1 M), snap-frozen instantly in liquid nitrogen, and stored at −80 °C for total RNA extraction. A portion of digestive gland tissue was cut and fixed in 4% paraformaldehyde (PFA) for fluorescent in situ hybridization (ISH).

### 2.3. Tissue Collection for mRNA Expression Analysis

#### 2.3.1. Collection of Different Organs from Pacific Abalone

After anesthetizing with 5% MgCl_2_, 15 abalones of both sexes were randomly sacrificed for different tissue sample collection including muscle (MUS), mantle (MA), gill (GIL), testis (TE), ovary (OV), hemocyte (HCY), heart (HRT), digestive gland (DG), cerebral ganglion (CG), and pleuropedal ganglion (PPG). Immediately after collection, tissue samples were carefully washed with 0.1 M PBS, snap-frozen in liquid nitrogen (LN), and stored at −80 °C for total RNA extraction.

#### 2.3.2. Collection of Samples at Different Embryonic and Larval Developmental Stages

Artificial fertilization of Pacific abalone was performed following the method described by Hossen et al. [22] and Sukhan et al. [23]. In brief, reproductively mature male and female abalones were induced by the traditional method of sunlight exposure (60 min in shell-side down and 40 min in shell-side up condition) and then male and female abalones were separately induced to release eggs and sperm separately by ultraviolet (UV)-treated seawater treatments. After spawning, unfertilized eggs and sperm were collected for further experiment and fertilization. About 30,000 eggs were mixed with sperm, maintaining a ratio of 1:10,000 for fertilization, and after 10 min, fertilized eggs were washed with filtered seawater. Successfully fertilized eggs were collected through subtle observation under the microscope. After every 10 min, samples were checked under a microscope and target stages (2-cell embryos, 4-cell embryos, blastula, trochophore, early veliger larvae, and late veliger) were collected. Immediately after collection, all samples were snap-frozen in LN and stored at –80 °C until total RNA extraction.

### 2.4. Total RNA Extraction and cDNA Synthesis

Extraction of total RNA from all tissue sample was performed using ISOSPIN Cell & Tissue RNA kit (Nippon Gene, Tokyo, Japan), following the manufacturer’s protocol. From extracted total RNA, first-stand cDNAs were synthesized using Superscript III First-stand cDNA Synthesis kit (Invitrogen, Waltham, MA, USA) and oligo(dt) primer (Sigma-Aldrich, St. Louis, MO, USA). 5′- and 3′-RACE cDNAs were synthesized from 1 μL of digestive gland total RNA using a SMARTer^®^ RACE 5′/3′ Kit (Takara Bio Inc., Shiga, Japan). 

### 2.5. Cloning and Sequencing of Hdh-MLGF

#### 2.5.1. Cloning of the Partial Sequence

Reverse transcription polymerase chain reaction (RT-PCR) was performed using digestive gland tissue cDNA template, a set of forward and reverse primers, and Phusion^®^ High-Fidelity DNA Polymerase (New England Biolabs Inc., Ipswich, MA, USA) to obtain a partial fragment sequence of the Hdh-MLGF gene. The forward and reverse primer was designed from a known mollusk-like growth factor nucleotide sequence of *H. discus discus* (GenBank accession no. EF103349.1). Primers used in this experiment has been presented in Table 1. A 20 μL volume of reaction mix containing 1 μL cDNA template, 4 μL of HF buffer, 2 μL of dNTP mix, 0.5 μL of DNA polymerase, 1 μL each of forward and reverse primer, and 10.5 μL of sterile distilled water was prepared to conduct RT-PCR. The thermal cycling condition used for the RT-PCR were: initial denaturation at 95 °C for 3 min, followed by 35 cycles denaturation for 30 s at 95 °C, 30 s annealing at 58 °C, 45 s extension at 72 °C and a final extension for 5 min at 72 °C. After completion, gel electrophoresis was performed for PCR products using agarose gel (1.2 agarose/100 mL) in gel chamber and produced band specific to primers amplicon length were purified using a PCR Clean-Up-System kit (Promega, Madison, WI, USA) and Wizard^®^ SV Gel. Then, purified PCR product were ligated into vector (pTOP Blunt V2) (Enzynomics, Daejeon, Korea) and transformed into competent cell (DH5α) (Enzynomics, Daejeon, Korea). Finally, the positive clones were purified using a Hybrid-QTM Plasmid Rapidprep mini kit (GeneAll, Seoul, Korea) and purified clones were sequenced at Macrogen (Seoul, Korea).

#### 2.5.2. Cloning of the RACE Sequence (5′ and 3′)

To obtain the full-length nucleotide sequence of *Hdh-MLGF*, 5′ and 3′-rapid amplification of cDNA ends (RACE) were performed using a SMARTer^®^ RACE 5′/3′ kit (Takara Bio Inc., Shiga, Japan). Gene specific 5′-RACE and 3′-RACE primer were prepared from the obtained partial sequence including a 15 bp overlap nucleotide (GATTACGCCAAGCTT) at the 5′ ends of the primer sequence (Table 1). 5′-RACE and 3′-RACE PCRs were performed using 2.5 μL of RACE specific cDNA (5′-RACE or 3′-RACE), 1 μL of RACE primer (5′-RACE or 3′-RACE), 1 μL of SeqAmp DNA polymerase, 5 μL of universal primer mix (UPM), 25 μL of SeqAmp buffer, and 15.5 μL of PCR-grade water. Maintaining thermal condition recommended in the kit 30 cycle touchdown PCR was carried out for both 5′-RACE and 3′-RACE. After completion, PCR products were subjected to gel electrophoresis (1.2 agarose) and produced band were purified using a NucleoSpin^®^ Gel and PCR clean-up kit (MACHEREY-NAGEL GmbH & Co. KG, Düren, Germany). Then, purified PCR product (both 5′-RACE and 3′-RACE) were ligated into vector (linearized pRACE vector), transformed into competent cells (Stellar competent cells), and the sequenced at Macrogen (Seoul, Korea) after purified positive clones as described previously to obtain both 5′-RACE and 3′-RACE sequences. At last, 5′-RACE sequence, initially cloned partial sequence, and 3′-RACE sequence were combined to obtain the complete sequence. 

### 2.6. Analysis of the Cloned Hdh-MLGF Sequence

The nucleotide and protein sequence of cloned Hdh-MLGF were analyzed using several online tools. From the full-length nucleotide sequence, the deduced amino acid sequence of the *Hdh-MLGF* cDNA was generated using the European Molecular Biology Open Software Suit (EMBOSS) transeq (http://www.ebi.ac.uk/Tools/st/emboss_transeq/; accessed on 5 June 2022) online tool. Protein encoding segment and open reading frame from the nucleotide sequence were predicted using ORF finder (https://www.ncbi.nlm.nih.gov/orffinder/; accessed on 5 June 2022). Hdh-MLGF protein homology was analyzed using the online tools Basic Local Alignment Search Tool (http://www.ncbi.nlm.nih.gov/BLAST/; accessed on 9 June 2022). The theoretical isoelectric point (pI) and molecular weight of Hdh-MLGF protein were computed using online tool ProtParam (https://web.expasy.org/protparam; accessed on 9 June 2022). The signal peptide of the protein was predicted using the online tools SignalP-6.0 (https://services.healthtech.dtu.dk/services.php?SignalP-6.0; accessed on 10 June 2022). The online tool C-I-TASSER available at protein structure and function predicting server (https://zhanggroup.org/C-I-TASSER; accessed on 17 June 2022) was used to predict Hdh-MLGF gene ontology. The motifs and functional domains of Hdh-MLGF protein sequence were identified using online tools Motif scan (http://myhits.isb-sib.ch/cgi203bin/motif_scan; accessed on 17 June 2022), and Simple modulear Architecture Research Tool (SMART) (http://smart.embl-heidelberg.de/; accessed on 17 June 2022). Multiple Em for Motif Elicitation (MEME) online tool (http://meme-suit.org/tools/meme; accessed on 18 June 2022) was used to discover conserved motifs in the Hdh-MLGF amino acid sequence. Hdh-MLGF and related proteins of ADGFs were aligned using an online service Clustal Omega (https://www.ebi.ac.uk/Tools/msa/clustalo/, accessed on 12 June 2022) maintained by EMBL’s European Bioinformatics Institute (EMBL-EBI). The multiple sequence alignment of proteins was further edited and visualized using Jalview (version 2.11.2.0) software [24].

### 2.7. Phylogenetic Analysis

Hdh-MLGF and other related protein sequences of ADGF family members were retrieved from the database available at NCBI and then aligned using multiple sequence alignment program Clustal Omega [24]. The phylogenetic tree was constructed for aligned protein sequence using MEGA (Molecular Evolutionary Genetics Analysis) software (v. 11) with neighbor joining algorithm.

### 2.8. Three-Dimensional Modeling of Hdh-MLGF

The predicted three-dimensional (3D) structure of Hdh-MLGF was constructed using previously mentioned online program C-I-TASSER for predicting protein structure. The 3D structure was farther visualized using the University of California, San Francisco (UCSF) ChimeraX software (v. 1.4) (https://www.rbvi.ucsf.edu/chimerax/, accessed on 20 June 2022).

### 2.9. Localization of Hdh-MLGF Using Fluorescence In Situ Hybridization 

#### 2.9.1. Riboprobe Synthesis

For localization of *Hdh-MLGF*, in situ hybridization was performed using the standard protocol previously described by Sukhan et al. [25]. Fluorescence mRNA probes (sense and anti-sense) were prepared through amplification of the *Hdh-MLGF* fragment sequence using sense and anti-sense primers and subcloned into the pGEM-T easy vector (Promega, Madison, WI, USA). Riboprobes of sense and anti-sense were denominated with fluorescein-12-UTP (Roche, Mannheim, Germany) using T7and SP6 RNA polymerase (Promega, Madison, WI, USA). The linear plasmid of *Hdh-MLGF* was prepared using 10 μg cDNA fragments with the restriction enzymes NcoI or SalI (Promega, Madison, WI, USA). Then, the mixer containing 1 μg of linearized plasmid DNA, 2.0 μL of T7 or SP6 RNA polymerase, 4.0 μL 5× optimized transcription buffer, 2.0 μL dithiothreitol (DTT, 100 mM), 2.0 μL fluorescein RNA labeling mix, 2.0 μL RNase inhibitor, and 7.0 μL Rnase-free water was incubated for 2 h at 37 °C. After incubation (transcription reaction), the linearized plasmid DNA template was digested with RNaseOut (0.5 μL) and DNase I (2.0 μL) at 37 °C for 15 min. The prepared riboprobes were purified by ethanol precipitation with 1 μL of yeast transfer RNA (Sigma-Aldrich, St. Louis, MO, USA) and finally stored at −80 °C.

#### 2.9.2. Frozen Tissue Sections Preparation

PFA-fixed digestive gland tissues of Pacific abalone were infiltered in 30% sucrose solution and embedded in optimum cutting temperature (OCT) compound (FSC 222, Leica Biosystems, Wetzlar, Germany). The embedded digestive gland tissues were then sectioned in transverse orientation at a thickness of 8μM using a cryostat device (CM 3050; Leica, Wetzlar, Germany). The sectioned tissues were collected onto SuperFrost@Plus slides (VWR International, Radnor, PA, USA) and immediately stored at −20 °C until further use in ISH.

#### 2.9.3. Fluorescence In Situ Hybridization (FISH)

FISH was performed following the protocol described by Sukhan et al. [25] and digoxigenin in situ hybridization manual with slight modification. In brief, a total volume of 50mL hybridization buffer was prepared using deionized formamide (25 mL), 20× saline sodium citrate (12.5 mL), 0.1% Tween-20 (0.5 mL) 1 M citric acid (0.46 mL) and diethyl pyrocarbonate (DEPC)-treated water (11.54 mL). The digestive gland tissue cryosections were prehybridized using yeast tRNA with hybridization buffer mix maintaining a ratio of 9:1 for 2 h at 65 °C, followed by hybridization with a fluorescein-12-UTP labeled RNA probe (200ng/mL) at 65 °C for overnight. Next, the hybridized tissue sections were sequentially washed with a degraded series (75%, 50% and 25%) of 2× SSC mixed hybridization buffer at 65 °C for 10 min each. Then, the tissue sections were cleansed with 0.2× SSC and 2× SSC for 15 min each. Afterward, the tissue sections were sequentially washed at room temperature with above mentioned degraded series of PBS and PBST mixed 0.2× SSC and then 5 min with PBST alone. The tissue sections were then incubated at room temperature for 1 h with calf serum (10%) and then incorporated with an antigen binding antibody, anti-digoxigenin-fluorescein (diluted 1:500 ratio in calf serum). After that, the tissue sections were washed for 10 min each with PBST at room temperature, followed by three times washing for 5 min each with alkaline Tris buffer. At last, counterstaining and mounting were performed using VECTASHIELD with 4′,6-diamidino-2 phenylindole (DAPI) (Vector Laboratories, Inc., Newark, CA, USA). The fluorescence signals of *Hdh-MLGF* were visualized and captured using a ZEISS LSM 900 with Airyscan 2 confocal microscope (ZEISS, Germany). Subsequently, the signal contained images were processed and visualized with ZEISS ZEN 3.2 software.

### 2.10. Semi-Quantitative Reverse Transcription-Polymerase Chain Reaction (RT-PCR)

To observe the expression pattern of Hdh-MLGF gene in different tissues of Pacific abalone, semi-quantitative RT-PCR was performed using gene-specific primers (forward and reverse). RT-PCR was performed with a reaction mixture (20 μL) containing a cDNA template (1 μL), forward and reverse primers (1 μL each), 2× Prime Taq premix (10 μL) (GENETBIO, Yuseong-gu, Korea), and ultra-pure water (7 μL). Partial sequence cloning PCR condition was maintained during incubation. The expression levels of *Hdh-MLGF* mRNA were observed in TE, OV, PPG, CG, GIL, DG, HRT, MA, MUS, and HCY of Pacific abalone. The β-actin gene of *H. discus hannai* (GenBank accession no. AY380809) was used as control due to its stable expression in Pacific abalone.

### 2.11. Quantitative Real-Time PCR (qRT-PCR) Analysis

For quantification of relative mRNA expression levels of *Hdh-MLGF* in Pacific abalone qRT-PCR analysis was performed in different tissues. *Hdh-MLGF* expression levels were observed in different organs of adult Pacific abalone as well as different embryonic and larval developmental stages of Pacific abalone.

All qRT-PCR assays were performed using a 2×qPCRBIO SyGreen Mix Lo-Rox kit (PCR Biosystems Ltd., London, UK) manual as described previously [23]. A total volume of 20 μL reaction mixture containing 1 μL cDNA template, 101 μL SyGreen Mix, 11 μL gene specific forward and reverse primer each, and 101 μL ultra-pure water was prepared to perform qRT-PCR in a LightCycler^®^ 96 System (Roche, Germany). Triplicate reactions were performed for *Hdh-MLGF* gene in each tissue sample. The conditions for PCR amplification were: initial incubation for 2 min at 95 °C, followed by 40 cycles of a three-step amplification 30 min at 95 °C, 20 s at 60 °C, and 30 s at 72 °C. The instrument default setting was maintained as melting temperature. Fluorescence reading was recorded for quantification at the end of each cycle. The relative mRNA expression was quantified using the 2^−ΔΔCT^ method with Pacific abalone *β-actin* gene as an internal reference. All primers used in qRT-PCR analysis are presented in Table 1.

### 2.12. Statistical Analysis

The mRNA expression values were statically analyzed and expressed as mean ± standard error of the mean (SEM). Changes in relative mRNA expression were analyzed by one-way analysis of variance (ANOVA) using GraphPad Prism 9.3.1 software. Statistical significance was set at *p* < 0.05. 

## 3. Results

### 3.1. Haliotis Discus Hannai MLGF Sequence

The full-length cDNA sequence of *Haliotid discus hannai* mollusk-like growth factor (MLGF) gene, referred to *Hdh-MLGF* in this study (GenBank accession no. ON803449) was successfully cloned from digestive gland tissue. The assembly of three cDNA fragments yielded a 1566 bp open reading frame (ORF), 48 bp 5′-untranslated region (UTR), and a 215 bp 3′-untranslated region (Figure 1). A putative polyadenylation signal (AATAAA) was found in its nucleotide sequence at 46 bp upstream of the poly-A tail. The *Hdh-MLGF* gene encodes a protein sequence with 521 deduced amino acid residues. The molecular weight and isoelectric point (pI) of the Hdh-MLGF protein were 60.14978 kDa and 5.78, respectively. The aliphatic index and instability index (II) of Hdh-MLGF were computed as 90.02 and 44.37, respectively.

### 3.2. Key Features of the Hdh-MLGF Amino Acid Sequence

Sequence analysis using SignalP-6.0 predicted a signal peptide containing 29 amino acids at positions 1M–29T in the N-terminal region of Hdh-MLGF protein (Figure 1). The Hdh-MLGF sequence had an ADA domain at the 175Q-497K amino acid position and an additional N-terminal ADA at position 12L-103I of the amino acid sequence. The sequence had four N-glycosylation sites at the 77N-80I, 128N-131Y, 184N-187I, and 379N-382R positions. A total of 11 casein kinase II phosphorylation sites with the structure [S/T]-X(2)-[D/E] were found in the amino acid sequence at positions 120T-123D, 198T-201D, 257S-260D, 267T-270D, 271T-274D, 344S-347D, 361T-364E, 366S-369D, 440S-443D, and 464T-467D. There were two N-myristoylation site structures found at positions 110G-115H and 116G-121S, as well as six protein kinase C (PKC) phosphorylation sites with the structure [ST]-X-[RK] at positions 90S-92K, 96S-98K, 130T-132R, 267T-269R, 380T-382R, and 489S-491K.

In the motif analysis, eight predicted motifs were found in the Hdh-MLGF protein sequence, which were completely homologous to the Hdd-MLGF motifs but were found to be asymmetrical when compared with other members of the ADGF subfamily (Figure 2). The length of all motifs was within 20 to 50 amino acids. Within the predicted eight motifs, three were detected from the N-terminal region and two were from the C-terminal region of the amino acid sequence.

In multiple sequence alignments, six cysteine residues were conserved in the MLGF amino acid sequence, among which one was conserved in the ADGF subfamily (Figure 3). Though ADA and an additional N-terminal ADA domain are the common features of the ADGF subfamily, multiple sequence alignments showed that the ADA domain was relatively conserved than the additional N-terminal ADA domain (Figure 3).

### 3.3. Homology Analysis

Amino-acid-based similarity analysis showed that the Pacific abalone, Hdh-MLGF shared the highest identity (98.65%) with the disc abalone Hdd-MLGF, whereas it showed the lowest identities with the sea slug (45.87%), zebrafish (41.94%), frog (41.52%), silkworm moth (35.92%), and fly (35.37%) (Table 2).

The percent similarity of Hdh-MLGF was found to be chronological with the percent identity. The maximum similarity of Hdh-MLGF was computed as 99.04% with Hdd-MLGF, whereas a minimum similarity of 47.43% was found with Gmm-TSGF-1 (Table 3).

### 3.4. Three-Dimensional Modeling of Hdh-MLGF

The three-dimensional structure constructed using C-I-TASSER was dominated by primary (linear) and secondary (beta-sheet and alpha-helix) structures. The result found that the predicted spatial structure of Pacific abalone Hdh-MLGF shared the most similarity with the structure of disc abalone Hdd-MLGF (Appendix A). Hdh-MLGF had four zinc ion binding residues at positions 113H, 115H, 357H, and 442D (Figure 4A), as well as four catalytic residues at positions 360E, 325G, 443D, and 385H (Figure 4B).

### 3.5. Gene Ontology Analysis

Amino acid sequence-based gene ontology (GO) analysis predicted that Hdh-MLGF mainly has ADA activity with GO confidence score 0.99. It also has growth factor activity (confidence score of 0.67) and a zinc ion binding (confidence score of 0.69) molecular function (Figure 5).

### 3.6. Phylogenetic Analysis

A phylogenetic tree was constructed based on multiple amino acid sequences to ascertain the relationship of the Pacific abalone MLGF protein sequence with other growth factors of vertebrates and invertebrates in the ADGF subfamily. The result revealed that *H. discus hannai* MLGF was closely related to *H. discus* MLGF and clustered in a separate cluster in mollusks, including *Aplysia californica* MDGF, with a 97-bootstrap value (Figure 6), suggesting that it was indeed the ortholog of the Ac-MDGF gene and reflecting the high conservation of this peptide sequence.

### 3.7. Hdh-MLGF Expression in Different Tissues of Pacific Abalone 

The relative mRNA expression levels of *Hdh-MLGF* in different tissues were measured by qRT-PCR. The expression of Hdh-MLGF in testis was regarded as reference value (value was set 1 for testis), and data was normalized. Tissue expression analysis revealed that the digestive gland showed significantly higher (*p* < 0.05) mRNA expression compared to other tested tissues. On the contrary, the lowest *Hdh-MLGF* expression was observed in the cerebral ganglion, heart, gill, mantle, and pleuropedal ganglion. Although some expression was found in the ovary, muscles, and hemocyte, it was also significantly lower than in the digestive gland (Figure 7). 

### 3.8. Tissue Specific Distribution of Hdh-MLGF

In tissue distribution analysis, the *Hdh-MLGF* gene was only expressed in the digestive gland tissue of Pacific abalone (Figure 8). Other organs such as the testis, ovary, ganglions, gill, heart, mantle, muscle, and hemocyte did not show *Hdh-MLGF* expression. The tissue distribution was compared with the *Hdh-β-Actin* gene, which was found to be abundantly expressed in all organs.

### 3.9. Localization of Hdh-MLGF in the Digestive Gland of Pacific Abalone 

FISH was performed to detect the *Hdh-MLGF* mRNA in the digestive gland of Pacific abalone. Confocal laser image scanning showed the antisense probe expressed positive signals in the submucosa layer of the digestive gland (Figure 9).

### 3.10. Expression of Hdh-MLGF at Different Embryonic Developmental Stages 

The results of qRT-PCR analysis confirmed that *Hdh-MLGF* mRNA was differentially expressed throughout the embryonic and larval developmental stages. After fertilization, the expression of *Hdh-MLGF* increased and peaked in the two-cell stage, which was the maximum expression level seen in cell division compared to the unfertilized egg. The lowest expression was found in the blastula stage as compared with the other stages. However, *Hdh-MLGF* was highly expressed in the trochophore and early veliger larvae (Figure 10), but significantly reduced at late veliger larvae stage.

## 4. Discussion

Growth factors play a key role in biophysical processes [26,27] including cell growth, cell division, migration, apoptosis, survival, wound healing, and metabolism. MLGF, a newly discovered secreted protein from abalone, is a member of the ADGF subfamily. In this study, the full-length cDNA sequence of the *MLGF* gene was isolated for the first time from the digestive gland tissue of Pacific abalone, *H. discus hannai*. The protein structure of the cloned *Hdh-MLGF* sequence possessed some key features common to the ADGF subfamily, including an N-terminal signal peptide, PKC phosphorylation sites, a conserved ADA domain, and an additional N-terminal ADA domain. A signal peptide at the N-terminal region of the amino acid sequence indicates that Hdh-MLGF is a secreted protein [28]. Potential phosphorylation sites detected in the cloned sequence might play an indispensable role in signal transduction pathways [19]. In addition, the unique N-terminal region of MLGF may have classic growth factor functions. It may bind to specific receptors on the target cell surface to stimulate a signaling cascade, and the ADA domain is necessary to modify another protein needed for the signal to be transmitted [29]. Most known growth factors function by binding to cell surface receptors and activating signal transduction pathways [30]. 

ADA activity have been reported in most of the ADGFs, including *H. discus hannai* MLGF, *A. californica* MDGF [29], *S. peregrina* ADGF-A [2], *L. longipalpis* ADA [31,32], and the *Drosophila* homologues, ADGF-A and ADGF-D. The ADA domain catalyzes the conversion of adenosine to inosine, thereby depleting the intra- and extra-cellular adenosine concentration [33]. In the Pacific abalone, Hdh-MLGF might stimulate cell growth through this indirect mechanism of converting adenosine to inosine (Figure 4B) by ADA, which is indispensable for the receptor to transmit the growth signal into the cells [2]. The previous findings suggests that adenosine to inosine conversion is necessary for mitogenic response [33]. 

The predicted 3D structure of Hdh-MLGF protein resembles the protein structure of disc abalone MLGF (Appendix A). The mitogenic activity of *Hdh-MLGF* requires catalytic activity, a common feature of the ADGF subfamily proteins [5], which might be regulated by the catalytic active site (Figure 4B). During catalysis, residue H385 (Histidine) directs the stereo-specific attack of water molecules in conjunction with residues D442 (Aspertic acid), as well as G325 (Glycine), and D443 (Aspertic acid) which form hydrogen bonds with nitrogen atoms in the purine ring; and residue E360 (Glutamic acid) is involved in the proton donating step (Figure 4B) [31,34]. There are also several loops (αα, αβ, αββ, αβα, etc.) present on the surface of MLGF that might be involved in growth factor activity.

In the phylogenetic tree, Hdh-MLGF clustered separately with Hdd-MLGF and Ac-MDGF, indicating a mollusk-specific cluster in the ADGF subfamily. This mollusk-specific invertebrate growth factor group is phylogenetically close to the vertebrate-specific growth factor, CECR. The protein identity and similarity results also support these findings. The conserved domain, sequence similarity, gene ontology, and phylogenetic analysis indicate that *Hdh-MLGF* is a member of the mollusk-specific ADGF family (MDGF).

Cellular localization of the *MLGF* mRNA transcript using in situ hybridization has not been reported yet in any molluscan species. However, in situ hybridization has previously been performed for other genes, including *CECR1* [2], *ADGF-A*, and *ADGF-D* [5] in the ADGF subfamily. In semi-quantitative RT-PCR, *Hdh-MLGF* only in digestive gland tissue of adult Pacific abalone. Moreover, fluorescence in situ localization expressed positive signal in the submucosa layer of the digestive tubule. As, Hdh-MLGF is a secreted protein and highly expressed in digestive gland tissue it may be the secretion of submucosal gland of Pacific abalone. Digestive gland secreted growth factors are mainly involved in gastrointestinal adaptation [35]. In situ hybridization localization of *MLGF* in the submucosa layer of the digestive tubules of Pacific abalone may be related to the gastrointestinal adaptation of adult abalone, as the interaction of nutrition and growth factors is poorly understood [35]. Different proteins, enzymes, and hormones have been reported to secrete from the digestive gland and digestive system. For instance, taste salivary growth factor, gastrin (growth factor), mucin, secretin, motilin, intrinsic factor, peptidase, Amylases, etc., from different vertebrates and invertebrates and play a role in food digestion. Digestive enzymes or proteins are involved in growth performance through nutrient metabolism. The hydrolase activity of Hdh-MLGF may acts on carbon–nitrogen bonds during pyrimidine biosynthesis.

The expression of *Hdh-MLGF* appears to be restricted to the digestive gland of adult Pacific abalone. In *Aplysia*, *MDGF* expression was found to be restricted to the atrial gland [28]. Even though previous studies have found ADGF subfamily members to be neurotrophic factors (involved in neuronal growth and survival) in juvenile and injured central nervous system (CNS) of adult *Aplysia* [9]. In goldfish, conversion of adenosine to inosine was also involved in axon outgrowth and the expression of growth-associated protein 43 kDa (GAP-43) [36]. However, the current study found negligible or no expression in the central nervous system (cerebral ganglion and pleuropedal ganglion) of adult abalone. As the present study performed *MLGF* mRNA expression analysis based on adult Pacific abalone organs, and *MLGF* has similar molecular functions and nature as *MDGF* (Appendix A), the *MLGF* gene may not be expressed in the central nervous system in adult abalone. This finding indicates that *Hdh-MLGF* expression may provide a stage-specific mechanism for elevating inosine levels that may create a permissive environment for growth and development.

During early development, a detectable level of mRNA expression suggests that *Hdh-MLGF* might be important for the embryonic and larval development of Pacific abalone. The expression of Hdh-MLGF was found from the unfertilized egg to larval development stages. The higher expression at the cellular level compared to the unfertilized egg indicates mitogenic activity of MLGF in Pacific abalone. In this stage, single cell become double. In blastula stage, shell formation initiates, and the lowest expression in this stage suggest that Hdh-MLGF have no function in shell development. Significantly higher expression in the trochophore and veliger larvae stages is a sign of *Hdh-MLGF* involvement in larval development. Formation of cilia, and larval movement is the key morphological change occurred in trochophore larvae stage. Higher expression of *Hdh-MLGF* mRNA is the indicative of higher production of adenosine and increases the growth factor activity that stimulates embryonic and larval cell growth through signal transduction pathways. However, the reason for the comparatively lower *Hdh-MLGF* expression in the late veliger larvae stage is unknown. Previously, Zurovec et al. [5] found higher expression in the embryonic mesoderm cell of *Drosophila*, indicating involvement of ADGF-A in embryonic development, whereas lack of ADGF-A caused changes in cell differentiation, fat body disintegration, and developmental delay [37]. In *Drosophila* larvae, ADGF-A and ADGF-D (both having ADA) strongly stimulate cell growth by depletion of extracellular adenosine through a catalytic process [11]. However, ADGF (ADGF-E) without an active ADA does not exhibit growth factor activity. Moreover, regulation of embryonic cell (NIH-Sape-4 cell) growth by growth factor MDGF in *Aplysia* and ADA in calf is also evident [13,29]. So, structural similarity, functional mechanism, and mRNA expression clearly indicate that *Hdh-MLGF* is involved in embryonic and larval development of Pacific abalone. 

## 5. Conclusions

The present study represents the first report describing the characterization and expression pattern of *MLGF* in Pacific abalone, *H. discus hannai*. *Hdh-MLGF* stimulates cell growth through indirect enzymatic (catalytic) activity, converting adenosine to inosine. The expression of *Hdh-MLGF* in the early developmental stages suggests that *Hdh-MLGF* is involved in the embryonic and larval development of Pacific abalone. However, mRNA expression in adult tissues and in situ localization indicate that *Hdh-MLGF* may also involve in gastrointestinal adaptation.

## Figures and Tables

**Figure 1 biology-11-01445-f001:**
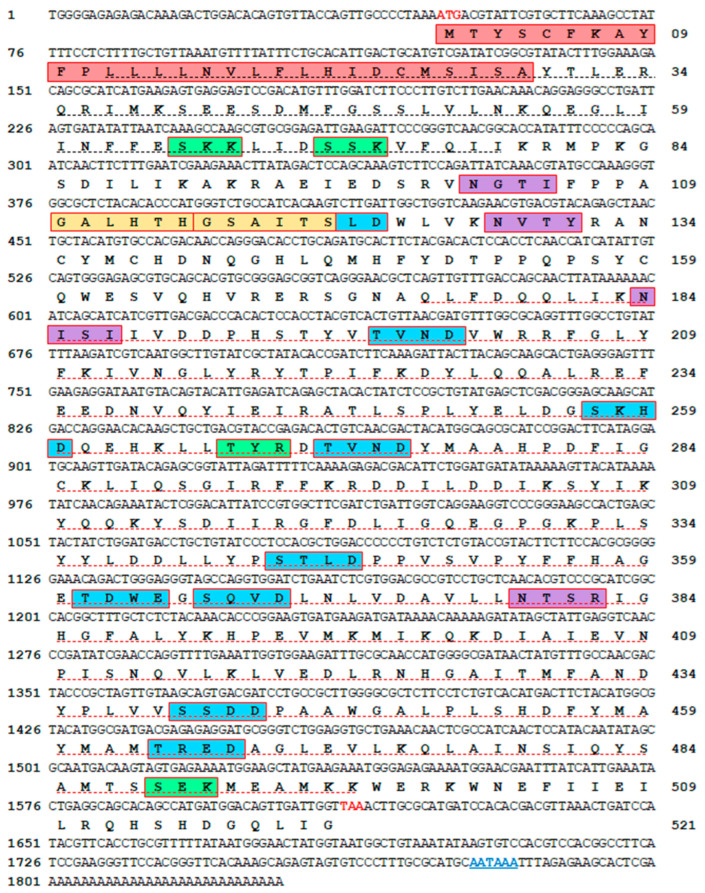
The Full-length nucleotide and amino acid sequences of *Hdh-MLGF* (GenBank accession no. ON803449). The numerical numbers at the left and right side of the sequence indicate the position of the nucleotide and amino acid sequence of the *Hdh-MLGF* gene, respectively. The coding regions starting with the start codon (ATG) and ending with the stop codon (TAA) are shown in red bold letters. The signal peptide for the secreted Hdh-MLGF protein has been indicated by the red box. The adenosine deaminase domain is marked with a red underline and an additional N-terminal adenosine deaminase with a black underline. N-glycosylation sites are denoted with a violet box. Potential protein kinase C (PKC) phosphorylation sites are pointed out with a green box. Predicted casein kinase II phosphorylation sites are marked with a blue box. The N-myristoylation site is boxed in yellow. The putative polyadenylation signal is denoted by a blue colored letter with a plain underline.

**Figure 2 biology-11-01445-f002:**
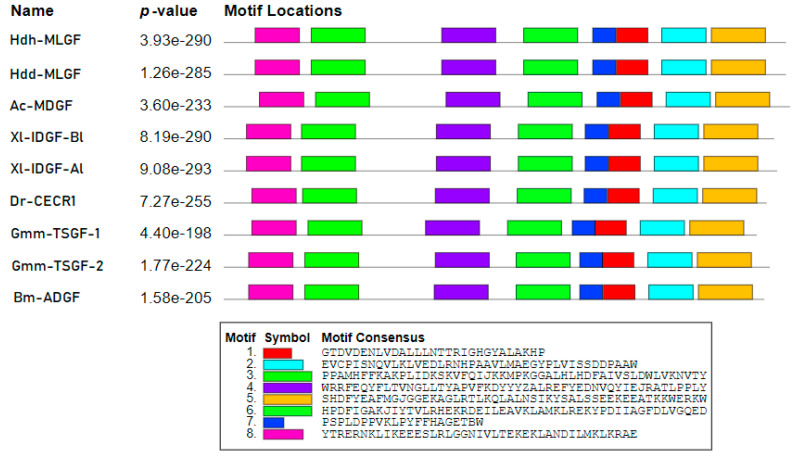
Schematic diagram of detected motifs in the Hdh-MLGF protein sequence and other ADGF subfamily members of vertebrates and invertebrates. Detected motifs are explained by different colors. The motif analysis included the Hdh-MLGF amino acid sequence and other growth factors of the following ADGF family members: MLGF of *H. discus* (ABO26607.1), MDGF of *Aplysia californica* (AAD13112.1), IDGF-Al of *Xenopus laevis* (AAY42596.1), IDGF-Bl of *Xenopus laevis* (AAY42597.1), CECR1 of *Danio rerio* (AAL40922.1), TSGF-1 precursor of *Glossina morsitans* (AAD52850.1), TSGF-2 of *Glossina morsitans* (AAD52851.1), and ADGF precursor of *Bombyx mori* (NP_001098698.1).

**Figure 3 biology-11-01445-f003:**
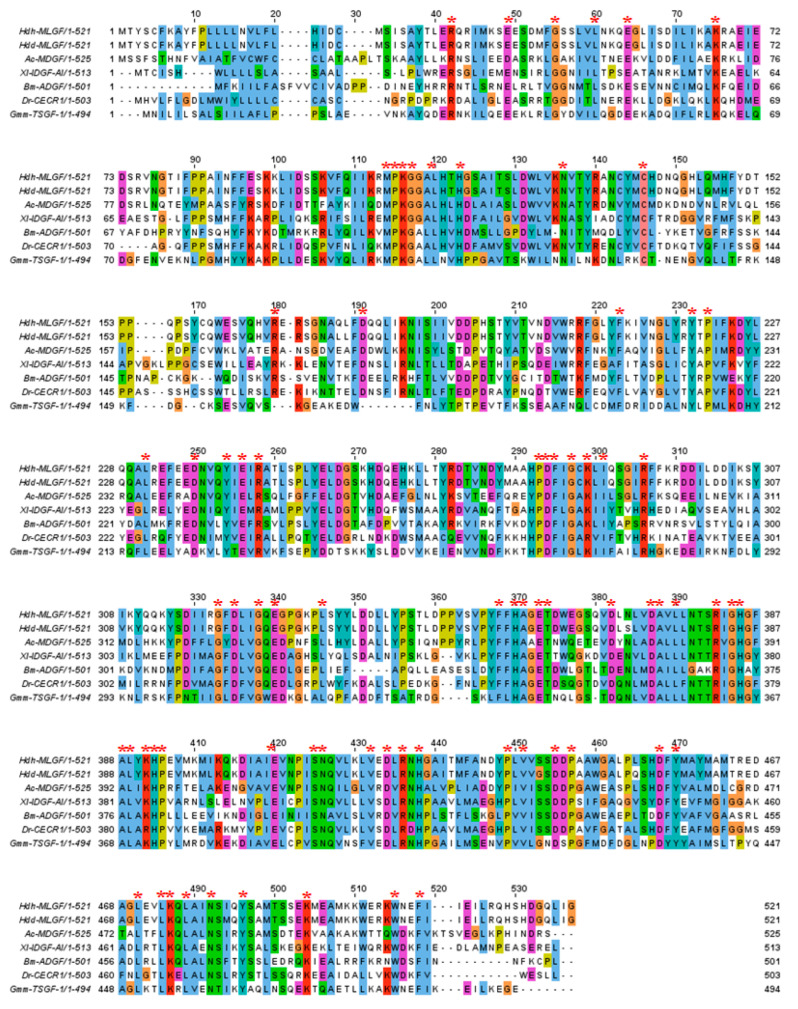
Multiple sequence alignments of mollusk-like growth factor from the deduced amino acid sequences of *H. discus hannai* (accession no. ON803449), *H. discus* (ABO26607.1), *Aplysia californica* (AAD13112.1), *Xenopus l*aevis (AAY42596.1), *Bombyx mori* (BAF73622.1), *Danio rerio* (AAL40922.1), and *Glossina morsitans* (AAD52850.1). Conserved residues are indicated by an asterisk.

**Figure 4 biology-11-01445-f004:**
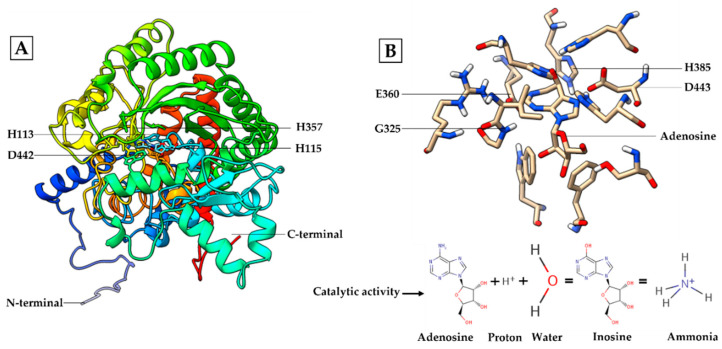
Three-dimensional structure and zinc ion binding sites prediction of Hdh-MLGF from the Pacific abalone, *H. discus hannai* (**A**) and catalytic activity with binding sites (**B**).

**Figure 5 biology-11-01445-f005:**
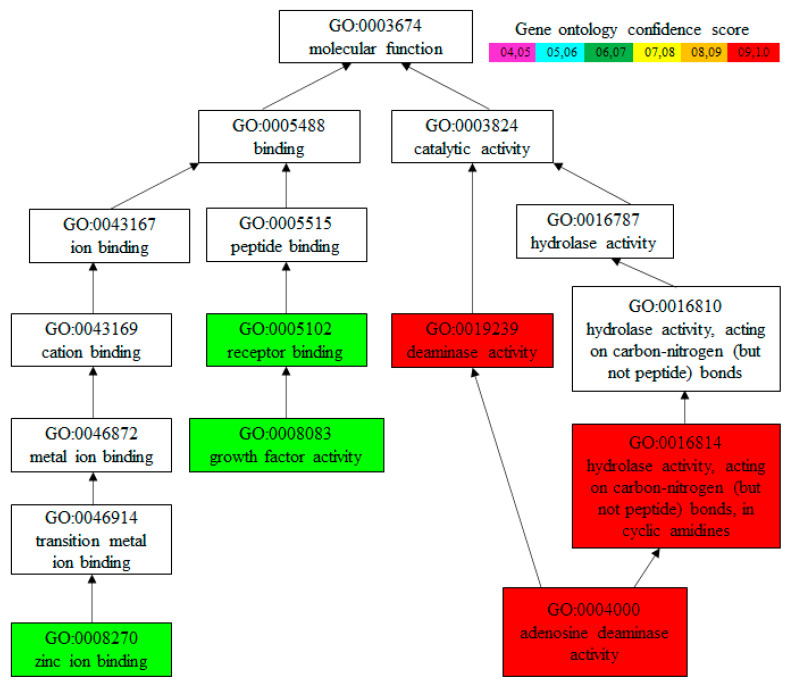
Functional analysis of the Hdh-MLGF amino acid sequence of Pacific abalone, *H. discus hannai*. Different color box in the upper side of the figure indicates the confidence score.

**Figure 6 biology-11-01445-f006:**
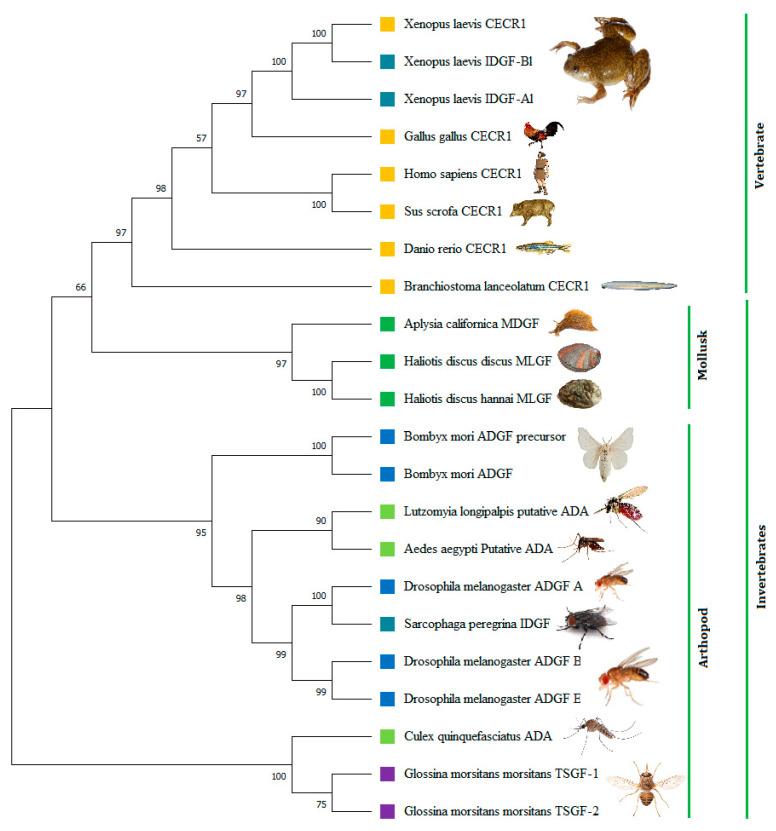
Phylogenetic tree constructed by the bootstrap neighbor-joining method after ClustalW alignment based on the amino acid sequences of different ADGF family members. The numbers at the nodes indicate bootstrap probability. Different colors of squares indicate different genes in the growth factor family ADGF, for instance: the orange square box for CECR1 (specific to vertebrate); green square for mollusks specific growth factor MDGF and MLGF; blue square for ADGF; teal square for IDGF (vertebrate and invertebrates separately formed cluster); lime square for ADA and violet square for TSGF. The sequences with their protein IDs used to construct the phylogenetic tree are as follows: MLGF of *H. discus hannai* (accession no. ON803449), *H. discus* (ABO26607.1), MDGF of *Aplysia californica* (AAD13112.1), IDGF of *Sarcophaga peregrina* (BAA11812.1), IDGF-Al of *Xenopus laevis* (AAY42596.1), IDGF-Bl of *Xenopus laevis* (AAY42597.1), CECR1 of *Xenopus laevis* (AAX10952.1), *Gallus* (AAX10953.1), *Homo sapiens* (CAG30303.1), *Sus scrofa* (AAL40921.1), *Danio rerio* (AAL40922.1), *Branchiostoma lanceolatum* (CAH1231807.1), ADA of *Lutzomyia longipalpis* (AAF78901.1), *Aedes aegypti* (AAL76033.1), *Culex quinquefasciatus* (AAK97208.1), TSGF-1 precursor of *Glossina morsitans* (AAD52850.1), TSGF-2 of *Glossina morsitans* (AAD52851.1), ADGF of *Bombyx mori* (BAF73622.1), ADGF precursor of *Bombyx mori* (NP_001098698.1), ADGF A of *Drosophila melanogaster* (AAL40913.1), ADGF B of *Drosophila melanogaster* (AAL40920.1), and ADGF E of *Drosophila melanogaster* (AAL40910.1).

**Figure 7 biology-11-01445-f007:**
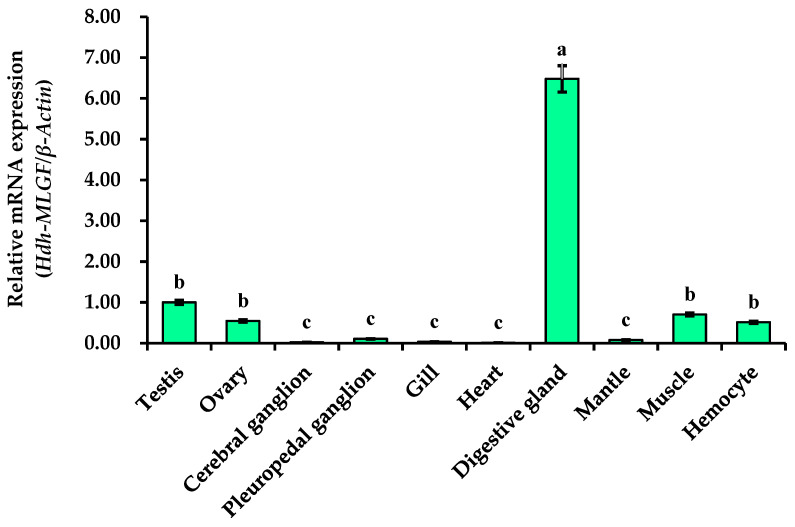
Expression of *Hdh-MLGF/**Hdh-β-Actin* mRNA in different tissues of Pacific abalone, *H. discus hannai.* In graph, error bars indicate standard error and different letters above the bar indicate significance difference (*p* < 0.05).

**Figure 8 biology-11-01445-f008:**
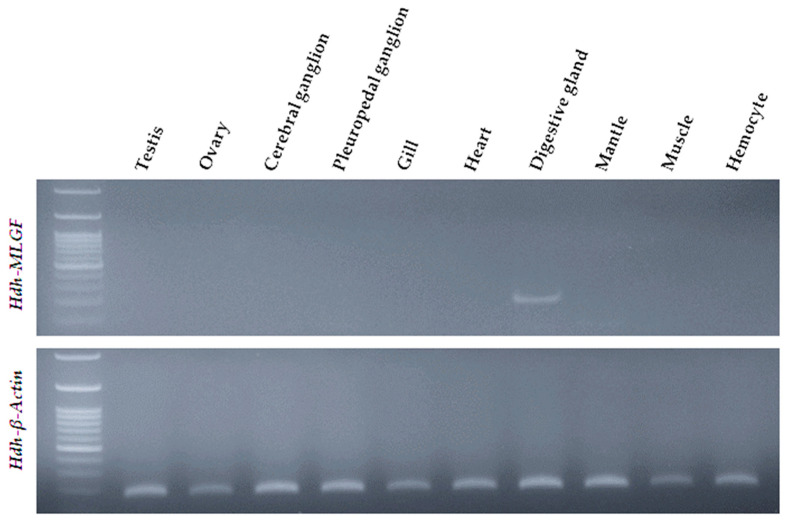
Tissue specific distribution of *Hdh-MLGF* and *Hdh-β-Actin* in the Pacific abalone, *H. discus hannai*.

**Figure 9 biology-11-01445-f009:**
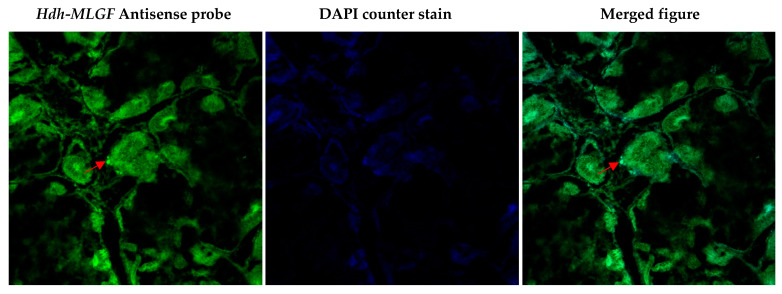
Microscopic (confocal laser scanning microscope) observation of *Hdh-MLGF* in digestive gland tissue of adult Pacific abalone using FISH.

**Figure 10 biology-11-01445-f010:**
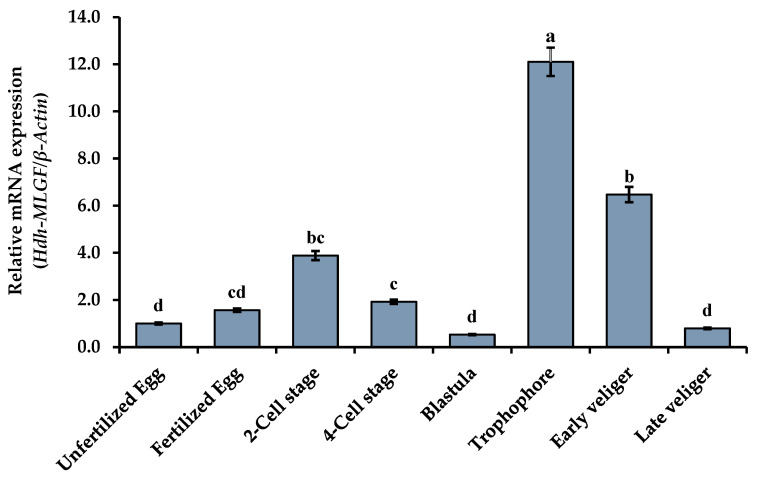
Expression of *Hdh-MLGF* at different stages of embryonic and larval developmental stages of Pacific abalone, *H. discus hannai.* In bar graph, error bars indicate standard error and different letters above the bar indicate significance difference (*p* < 0.05).

**Table 1 biology-11-01445-t001:** Different primers used for cDNA synthesis, cloning, expression analysis, and in situ hybridization.

Primer Name	Nucleotide Sequences	Purpose
Oligo dT (OdT)	GGCCACGCGTCGACTAGTACTTTTTTTTTTTTTTTTT	cDNA synthesis
Oligo dT adapter (AP)	GGCCACGCGTCGACTAGTAC	
MLGF-Fw	GAGAGACAAAGACTGGACAC	RT-PCR
MLGF-Rv	GTCAAACAACTGAGCGTTCC	
Hdh-MLGF 5′	GATTACGCCAAGCTTGTGTCTCGGTACGTCAGCAGCTTGTG	RACE PCR
Hdh-MLGF 3′	GATTACGCCAAGCTTCACAAGCTGCTGACGTACCGAGACAC	
Hdh-MLGF-ORF-Fw	CAGTTGCCCCTAAAATGACG	Full-length sequencing
Hdh-MLGF-ORF-Rv	CAGGTGAACGTATGGATCAG	
Hdh-MLGF-sense	CAGTTGCCCCTAAAATGACG	ISH
Hdh-MLGF-antisense	GGATCATGCGCAAGTTTAACC	
Hdh-MLGF-qRT-Fw	GAGCTAACTGCTACATGTGC	qRT-PCR
Hdh-MLGF-qRT-Rv	CGTCAACGATGATGCTGATG	
Hdh-β-Actin-Fw	CCGTGAAAAGATGACCCAGA	qRT-PCR
Hdh-β-Actin-Rv	TACGACCGGAAGCGTACAGA	

**Table 2 biology-11-01445-t002:** Amino acid percent identity of Hdh-MLGF compared with other members in the ADGF subfamily.

	Hdh-MLGF	Hdd-MLGF	Ac-MDGF	Xl-IDGF	Bm-ADGF	Dr-CECR1	Gmm-TSGF-1
Hdh-MLGF	100%						
Hdd-MLGF	98.65%	100%					
Ac-MDGF	45.87%	45.1%	100%				
Xl-IDGF	41.52%	40.74%	39.37%	100%			
Bm-ADGF	35.92%	35.32%	37.52%	39.12%	100%		
Dr-CECR1	41.94%	41.35%	39.76%	53.87%	37.72%	100%	
Gmm-TSGF-1	35.37%	34.58%	34.58%	38.73%	35.12%	37.17%	100%

**Table 3 biology-11-01445-t003:** Amino acid percent similarity of Hdh-MLGF compared with other members in the ADGF subfamily.

	Hdh-MLGF	Hdd-MLGF	Ac-MDGF	Xl-IDGF	Bm-ADGF	Dr-CECR1	Gmm-TSGF-1
Hdh-MLGF	100%						
Hdd-MLGF	99.04%	100%					
Ac-MDGF	57.38%	56.62%	100%				
Xl-IDGF	52.43%	51.65%	51.07%	100%			
Bm-ADGF	47.5%	46.9%	48.5%	50.09%	100%		
Dr-CECR1	52.48%	51.68%	51.29%	62.22%	48.9%	100%	
Gmm-TSGF-1	47.43%	46.64%	45.84%	47.82%	44.91%	47.11%	100%

## Data Availability

All data generated in this study are included in this article.

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
