# Peer review of "Characterization and Expression Analysis of Mollusk-like Growth Factor: A Secreted Protein Involved in Pacific Abalone Embryonic and Larval Development"

_biology, 2022, doi:10.3390/biology11101445_

Round 1
Reviewer 1 Report
Authors cloned a mollusk like growth factor from Pacific abalone, and the results were interesting.
1.The results should first show the expression level of the gene in different tissues, and then show the results of the gene in situ hybridizationï¼›
2.For the specifically expressed tissue lymph nodes, the article did not introduce and discuss in detail, such as the role of digestive gland, what other proteins are secreted, and what is the correlation significance with growth factor; If no other hormones or proteins have been reported to be expressed in digestive gland, please explain the significance of this report for our understanding of mollusc lymph nodes.
3. For the period of specific expression, there is no detailed introduction and discussion, such as what morphological characteristics of this period, or whether there are any landmark events. What events or morphological changes can a growth factor be associated with?
Author Response
Dear Editor/Reviewer,
Thank you very much for reviewing our manuscript. We also greatly appreciate the reviewers for their complimentary comments and suggestions. We have revised and corrected the manuscript accordingly, as suggested by reviewers.
Please find attached a point-by-point response (track changed) to the reviewer’s comments.
We profusely thank all reviewers and editor for sparing their valuable time to critically reviewing the manuscript and suggesting pertinent and very useful comments in order to improve the quality of our manuscript.
Reviewer 1 comments
- The results should first show the expression level of the gene in different tissues, and then show the results of the gene in situ hybridization.
Response: In the result part, after the expression level of the gene in different tissues, localization of Hdh-MLGF gene using in situ hybridization has been presented.
- For the specifically expressed tissue lymph nodes, the article did not introduce and discuss in detail, such as the role of digestive gland, what other proteins are secreted, and what is the correlation significance with growth factor; If no other hormones or proteins have been reported to be expressed in digestive gland, please explain the significance of this report for our understanding of mollusc lymph nodes.
Response: Different proteins, enzymes, and hormones have been reported to be secreted from the digestive gland and digestive system. For instance, tsetse salivary growth factor, gastrin (growth factor), mucin, secretin, motilin, intrinsic factor, peptidase, Amylases, etc. from different vertebrates and invertebrates and play a role in food digestion. Digestive enzymes or proteins are involved in growth performance through nutrient metabolism. The hydrolase activity of Hdh-MLGF acts on carbon-nitrogen bonds during pyrimidine biosynthesis.
- For the period of specific expression, there is no detailed introduction and discussion, such as what morphological characteristics of this period, or whether there are any landmark events. What events or morphological changes can a growth factor be associated with?
Response: The morphological changes involved in the period of specific expression have been discussed in the discussion part. During specific expression, morphological changes occurred in the embryonic and larval development stages of Pacific abalone, such as increased cell number (2 cell stage), started shell formation (blastula), cilia formation, and started movement (trochophore larvae). Growth factors are generally involved in cell division (mitogenic activity), cell growth, and cell proliferation. Hdh-MLGF is a member of the ADGF family, and most members of this family are involved in embryogenesis (embryonic and larval development).

Reviewer 2 Report
Comments to authors
The manuscript entitled “Characterization and expression analysis of Mollusk like growth factor: A secreted protein involved in Pacific abalone embryonic and larval development” deals with the description and expression of the mollusk like growth factor of the Pacific abalone, and its relationship with other vertebrate and invertebrate growth factors. The manuscript is well written, the analyses and results are very complete and the figures clearly summarize the main results. This paper provides interesting and valuable contributions, shedding light on the functioning of growth factors in mollusks and their role in embryonic and larval development. I think the manuscript will be of the interest of the readers of Biology. I only have a few minor comments (see below).
Minor comments
P. 2, L. 64: Please, change H. for Haliotis, as it is the first time you name it in the main text
P. 2, L. 70: Please, delete the comma after “United States”
P. 2, L. 71: Please, change the sentence as follows: “In recent decades, the natural abalone has (…), due to a high market demand, which resulted in…”
P. 2, L. 73: Change “thus” for “as a consequence” or “as a result”
P. 2, L. 75: there is a spelling mistake, please separate “the” from “available”
P. 2, L. 79: add “the” before “seed”
P. 2, L. 80: Please, change the sentence as follows: “Embryonic and larval development are (…) that can affect…”
P. 2, L. 80: Please, specify the species of abalone here and add “from abalone sea cage culture area of Wando-gun (South Korea), having…”
P. 2, L. 94: add “the” before “abalone”, and change “abalone” for “abalones”
P. 3, L. 106: change the sentence as follows: “We randomly selected 15 abalones of both secez that were anesthetized with 5% MgCl2 and then sacrificed for different tissue…”
P. 3, L. 115: change “abalone” for “abalones”
P. 14, Figure 6: You should specify in the figure caption the meaning of the different colours of the squares.
P. 16, L. 460-465: Did you check if these data met the assumption of homogeneity of variances (for example, with the test of Levene) that is needed for an ANOVA test?
P. 17, L. 487 and 503: add “the” before “Pacific abalone”
P. 18. L. 529-532: Please, rephrase this sentence, it is a bit confusing. Add “the” before “digestive gland”
P. 18, L. 548: I think you wanted to say “indicates” instead of indicating
P. 18, L. 557: please, change “the indication” for “an indicative”
Author Response
Dear Editor/Reviewer,
Thank you very much for reviewing our manuscript. We also greatly appreciate the reviewers for their complimentary comments and suggestions. We have revised and corrected the manuscript accordingly, as suggested by reviewers.
Please find attached a point-by-point response (track changed) to the reviewer’s comments.
We profusely thank all reviewers and editor for sparing their valuable time to critically reviewing the manuscript and suggesting pertinent and very useful comments in order to improve the quality of our manuscript.
Reviewer 2 comments
Comment: P. 2, L. 64: Please, change H. for Haliotis, as it is the first time you name it in the main text
Response: “H.” has been changed to “Haliotis” due to first time mentioned this name in the manuscript.
Comment: P. 2, L. 70: Please, delete the comma after “United States”
Response: After “United States” comma has been deleted in the text.
Comment: P. 2, L. 71: Please, change the sentence as follows: “In recent decades, the natural abalone has (…), due to a high market demand, which resulted in…”
Response: According to suggestion the sentence has been changed to “In recent decades, the natural abalone has experienced intensive fishing throughout the world, due to a high market demand, which resulted in natural stock depletion in most countries.
Comment: P. 2, L. 73: Change “thus” for “as a consequence” or “as a result”
Response: In line 73, “Thus” has been changed to “As a result”.
Comment: P. 2, L. 75: There is a spelling mistake, please separate “the” from “available”
Response: The attached word “theavailable” has been separated as “the available”
Comment: P. 2, L. 79: add “the” before “seed”
Response: The proposed correction has been made in the text.
Comment: P. 2, L. 80: Please, change the sentence as follows: “Embryonic and larval development are (…) that can affect…”
Response: The sentence has been changed to “Embryonic and larval development are essential processes that can affect the quality of seed and growth status as well as determine aquaculture economic performance.”
Comment: P. 2, L. 80: Please, specify the species of abalone here and add “from abalone sea cage culture area of Wando-gun (South Korea), having…”
Response: Abalone species has been specified as Pacific abalone in the text.
Comment: P. 2, L. 94: add “the” before “abalone”, and change “abalone” for “abalones”
Response: Recommended corrections has been done in the text.
Comment: P. 3, L. 106: change the sentence as follows: “We randomly selected 15 abalones of both secez that were anesthetized with 5% MgCl2 and then sacrificed for different tissue…”
Response: The sentence has been changed as “We randomly selected 15 abalones of both sexes that were anesthetized with 5% MgCl2 and then sacrificed for different tissue collection to perform cloning and isolation of MLGF.”
Comment: P. 3, L. 115: change “abalone” for “abalones”
Response: “abalone” has been changed to “abalones” in the text.
Comment: P. 14, Figure 6: You should specify in the figure caption the meaning of the different colours of the squares.
Response: The meaning of different colours of the squares has been specified in the figure caption as “Different coloures of squares indicate different genes in the growth factor family ADGF, for in-stance; orange square box for CECR1 (specific to vertebrate); green square for mollusks specific growth factor MDGF and MLGF; blue square for ADGF; teal square for IDGF (vertebrate and in-vertebrates separately formed cluster); lime square for ADA and violet square for TSGF.”
Comment: P. 16, L. 460-465: Did you check if these data met the assumption of homogeneity of variances (for example, with the test of Levene) that is needed for an ANOVA test?
Response: Yes, the data were checked for Homogeneity of variances in ANOVA test.
Comment: P. 17, L. 487 and 503: add “the” before “Pacific abalone”
Response: Recommended correction have been made in the text.
Comment: P. 18. L. 529-532: Please, rephrase this sentence, it is a bit confusing. Add “the” before “digestive gland”.
Response: The sentence covering line 529-532 have been rephrased and “the” has been added before ‘digestive gland’.
Comment: P. 18, L. 548: I think you wanted to say “indicates” instead of indicating
Response: Required correction has been made in the text.
Comment: P. 18, L. 557: please, change “the indication” for “an indicative”
Response: “the indication” has been changed to “an indicative” in the text.
